# The Release Behavior of Anthraquinones Encapsulated into Casein Micelles during In Vitro Digestion

**DOI:** 10.3390/foods12152844

**Published:** 2023-07-27

**Authors:** Uzma Sadiq, Fatima Shahid, Harsharn Gill, Jayani Chandrapala

**Affiliations:** 1School of Science, RMIT University, Bundoora, Melbourne, VIC 3083, Australia; uzma.sadiq@student.rmit.edu.au (U.S.); harsharn.gill@rmit.edu.au (H.G.); 2School of Chemistry, Monash University, Clayton, VIC 3800, Australia

**Keywords:** in vitro digestion, casein micelles, spray-dried microcapsules, anthraquinones, aloe vera plant, release behavior

## Abstract

The degradation of anthraquinones extracted from aloe vera plants can be prevented by encapsulating them in casein micelles (CMs). The oral, gastric, and intestinal digestion behavior of spray-dried microcapsules of casein micelles loaded with aloe vera-extracted anthraquinone powder (CMAQP), freeze-dried powder (CMFDP), and whole-leaf aloe vera gel (CMWLAG) obtained through ultrasonication was investigated. The results found that CMAQP and CMFDP dissolved slowly and coagulated into large curds during gastric digestion, improving the retention of anthraquinones in the digestive tract. In contrast, CMWLAG structure was destroyed and increased amounts of anthraquinones were released during oral and gastric digestion phases, indicating increased amounts of surface anthraquinones instead of the encapsulation of anthraquinones in the interior of CMs. The strong hydrophobic interactions protected anthraquinones within the core of CM for CMAQP and delayed diffusion. However, during SIF digestion, both CMAQP and CMFDP released significant amounts of anthraquinones, although CMAQP showed a much more controlled release for both aloin and aloe-emodin over SIF digestion time. The release behavior of anthraquinones from CM microcapsules was a function of the type of anthraquinone that was used to encapsulate. The present study provides insight into the release behavior of loaded bioactive compounds using food-grade CMs as the wall material during in vitro digestion and highlights the importance of the type of bioactive component form that will be encapsulated.

## 1. Introduction

Aloe vera is a popular plant known for its therapeutic properties, and it contains several bioactive compounds, including anthraquinones. Anthraquinones are a class of naturally occurring compounds found in plants that have been used for medicinal purposes for centuries [1]. These compounds have been reported to possess various biological activities, including antimicrobial [2], anti-inflammatory, and anticancer properties [3]. However, the bioavailability of anthraquinones in aloe vera is limited due to their rapid degradation after harvesting and their sensitivity towards food processing conditions [4]. 

One approach to overcome these limitations is encapsulating anthraquinones in a delivery system, such as casein micelles [5]. Casein micelles, the major protein component of milk, have been reported to enhance the bioavailability of hydrophobic compounds by encapsulating them in their internal structure [6]. Casein micelles have been extensively studied as potential delivery systems for benzydamine [7], vitamin C [8], emodin [9], vitamin D_2_ [10], and aloe vera-extracted anthraquinones [5]. Although our previous studies showed better encapsulation efficiencies and stabilities of spray-dried casein micelles loaded with anthraquinones [5], their ultimate health impacts still mainly depend on their bioavailability during in vitro digestion. 

In vitro digestion models are frequently exploited to examine the release behavior of blueberry anthocyanins [11], caffeine [12], curcumin [13], and Alfuzosin Hydrochloride that have been encapsulated into casein micelles. A casein micelle is a colloidal calcium phosphate (CCP)-mediated tight structure that frequently demonstrates distinctive stomach curd behavior and instantly hydrolyzes while in the intestine [14]. As the human stomach pH is profoundly acidic yet neutral in the duodenum, the casein micelles’ pH-dependent actions can be favorable for the controlled release of oral bioactive components. Thus, pH-sensitive casein micelle gel inflammatory behavior benefits the eternal release of bioactive components [15]. Previous studies showed that casein micelles generally aggregate via the introduction of acids and enzymes in the stomach [16] and thereby undergo structural changes due to digestion [17]. The most recent study [18] reported that the casein micelles are an encapsulating agent for anthocyanins by adding a chelating agent, dextran sulfate, to produce ACN-MC-DS ternary complexes. These complexes showed increased encapsulation efficiencies, and a reduced curdling ability resulted in higher gastric release in the stomach based on the CCP dissolution. Another study evaluated [19] the effect of temperature on pepsin-induced coagulation of milk casein at various temperatures and reported that casein micelles at 4 °C have looser and soft structures that could easily be hydrolyzed by the pepsin enzyme as opposed to body temperature (37 °C). Hence, the casein micelles’ stability, solubility, bioavailability, and release of anthraquinones during digestion are dependent on various factors, including temperature, gastric emptying time, dissolution degree of CCP, and finally the type of bioactive compound encapsulated [17]. 

Thus, the present study investigated the oral, gastric, and intestinal digestion of spray-dried microcapsules of casein micelles loaded with anthraquinone powder (CMAQP), freeze-dried powder of aloe vera (CMFDP), and whole-leaf aloe vera gel (CMWLAG). The release behavior through HPLC and structural and binding interaction changes through SEM and FTIR were investigated. 

## 2. Materials and Methods

### 2.1. Materials 

Aloe vera (*Aloe barbadensis*) leaves were collected from Aloe vera Australia (Goodman International, Brisbane, QLD, Australia). Casein micelle powder was purchased from Sigma Aldrich Pty Ltd. (Castle Hill, NSW, Australia). Pepsin from procin gastric mucosa, activity ≥32,000 Units/mg ((P6887) CAS No: 9001-75-6), α-amylase from human saliva, activity 1000–3000 Units/mg ((A0521), Lot# SLBX2676), Trypsin from porcin pancreas, activity 1000–2000 Units/mg ((T4799) CAS-No: 9002-07-7) were purchased from Sigma Aldrich Pty Ltd. (Castle Hill, NSW, Australia). HPLC-grade methanol, formic acid, and bile salts were also acquired from Sigma Aldrich Pty Ltd. (Castle Hill, NSW, Australia). Other chemicals were of analytical grade and obtained from Sigma Aldrich Pty Ltd. (Castle Hill, NSW, Australia). MilliQ water was used at all times. 

### 2.2. Methods

#### Ultrasonic-Assisted Encapsulation of Anthraquinones into CM and Preparation of Spray-Dried Microcapsules

Aloe vera leaves were cut into slices, blended, centrifuged, and filtered through vacuum filtration. Three parts of the liquid gel were divided to produce anthraquinone powders (AQPs), freeze-dried powders (FDPs), and whole-leaf aloe vera gels (WLAGs), as previously reported [4]. A 2% (*w*/*w*) casein micelle solution was prepared by dissolving calculated amounts of casein micelle powder in MilliQ water and vigorously shaking the solution for 30 s. A magnetic stirrer (9 MR Hei-Tec Stirrer Pt1000 V4 A) was used to stir the solution continuously for one hour (300 rpm) at a constant temperature of 50 °C [20].

The AQP, FDP, and WLAG samples prepared in the previous section were accurately measured and mixed with casein micelle solutions to produce concentrations of 20 mg/mL, 20 mg/mL, and 4 g/mL, respectively. This was then followed by ultrasound-assisted encapsulation under the conditions mentioned previously [5] using a 20 kHz ultrasound machine (500 W) operating at 50% amplitude, resulting in an actual power delivery of 39.74 W. The microcapsules of CMAQP, CMFDP, and CMWLAG were spray-dried using a mini spray dryer B-290 (BÜCHI Labor Technik AG, Meierseggstrasse 40 Postfach, Flawil, Switzerland) operating at predetermined inlet and outlet temperatures, as described in the previous article [20]

### 2.3. In Vitro Static Digestion 

The simulated salivary fluids (SSFs), simulated gastric fluid (SGF), and simulated intestinal fluids (SIFs) were prepared freshly by using stock electrolyte solutions, enzymes, CaCl_2_, and water, as described by [21,22] with slight modifications. The oral, gastric, and intestinal fluids were prepared as given in Table 1.

**SSF:** A total of 2 mL of SSF (Table 1) in 15 mL tubes was preheated at 37 °C for 10 min. A total of 2 g powdered microcapsules of casein micelles (controlled CMs), anthraquinone powder-loaded casein micelles (CMAQPs), freeze-dried powder-loaded casein micelles (CMFDPs), and whole-leaf aloe vera gel-loaded casein micelles (CMWLAGs), extracted anthraquinones powder (AQP) or freeze-dried powder (FDP) were added into the tube to create a thin paste-like consistency. A total of 10 µL of CaCl_2_ (0.3 M) and 100 µL of α-amylase (dissolved in SSF (50 µL/mL is equivalent to 75 U/mL)) were added and allowed to shake for 2 min. After this, 0.5 g samples were removed from each tube and snap freezing was carried out using dry ice.

**SGF**: The remaining bolus (2.5 g) was further used for gastric digestion. For this, 2.5 mL of the sample was mixed with SGF (2.5 mL) and pH was adjusted to 3.0 with 1 M HCL. The digesta was incubated using a shaking water bath TW-20 (JULABO, Marcon Boulevard, Allentown, PA, USA) with a stirring speed of 200 rpm at 37 °C and a significant amount of pepsin (2.6 mg) was rapidly added at a final activity of 2000 U/mL. Gastric digestion was allowed for 2 hours and 0.5 g of digesta was collected at 60 min and 120 min. At the end of the gastric digestion, the pH of chyme was adjusted to 7.0 by adding 1 M NaOH. 

**SIF**: Simulated intestinal digestion started by adding 4 mL of SIF, trypsin (0.36 mg), and bile salts (3 mL) to the pH-adjusted gastric chyme at a concentration of 100 U/mL and 10 mM, respectively. Intestinal digestion was conducted for 2 h at the same stirring speed of 200 rpm and 37 °C. Samples of the intestinal mixture were collected at 60 and 120 min. 

#### 2.3.1. Release Behavior of Anthraquinones during In Vitro Digestion

All the obtained samples from oral, gastric, and intestinal digestions were placed on dry ice to snap freeze and anhydrous methanol was added to stop the enzymatic activity using the method of [23] with some modifications. After that, samples were centrifuged (centrifuge 3–30 kHS-Sigma) at 19,000 rpm for 10 min to obtain the supernatant. Anthraquinones were extracted from the supernatant in methanol using an ultrasonication bath (FXP12, Unisonics, Brookvale, NSW, Australia) at ~4 ± 1 °C for 10 min. The anthraquinone contents in the supernatant were measured by HPLC as previously described [4]. Briefly, all samples were filtered by a 0.45 µ membrane filter before HPLC analysis [4]. Standard solutions with concentrations of 5 ppm, 10 ppm, 20 ppm, and 40 ppm for aloin, aloe-emodin, and rhein were prepared as previously reported [4]. An Agilent series 1250 infinity gradient HPLC (Agilent Technologies, Santa Clara, CA, USA) outfitted with a 600 solvent pump and a C18 reversed-phase packing column (Phenomenex XB-C18, 250 mm 4.60 mm, 3.6 m Aeris) was used to conduct the HPLC measurements. A binary mobile phase made up of water with 1 percent formic acid (A) and methanol (B) was used for gradient elution at 0.7 mL/min. The injection volume was 20 µL, and the elution was observed at 254 nm using a UV-Vis diode array detector [4]. The A.Q. contents in the supernatants, which represented A.Q.’s release, were measured by comparing the standards and the samples. The cumulative release was calculated by the equation given below.
Cummulative Release percentage(%)=Extracted AQ in the supernatant/Total anthraquinones added×100

#### 2.3.2. Scanning Electron Microscopy during In Vitro Digestion

Samples obtained during each phase of in vitro digestion were centrifuged at 19,000 rpm for 10 min using a centrifuge (Sigma centrifuge machine 3–30KS John Morris group, Whitehorse Road, Deepdene, VIC 3103, Australia) to recover the suspension. The suspension was pre-frozen at −80 °C and dried with a freeze dryer (Labconco FreeZone Triad Benchtop Freeze Dryer, Kansas City, MO, USA). The freeze-dried powder was evaluated for microstructure using a high-resolution scanning electron microscope (SEM) FEI Nova Nano 450 FEG-SEM at Monash Centre for Electron Microscopy (MCEM). Each sample was placed on the conductive adhesive and coated with graphite. The images were obtained at magnifications of 5000× and 10,000×. The acceleration voltage was kept between 5 and 15 KV. 

#### 2.3.3. Fourier Transform Infrared (FTIR) Spectroscopy during In Vitro Digestion

All obtained samples of CM controlled, CMAQP, CMFDP, and CMWLAG were freeze-dried after each digestion phase (oral, gastric, and intestinal) and were evaluated for structural modifications by Fourier transform infrared spectroscopy. FTIR spectra of the powdered samples were obtained using FTIR (Spectrum two, Perkin Elmer, North Ryde, NSW, Australia) furnished with IRWinLab FTIR software. Measurements were taken at 400–4000 cm^−1^ for every sample. Sixteen scans were performed, and the resolution used was 4 cm^−1^.

#### 2.3.4. Statistical Analysis

All measurements were performed at least in triplicates, and results were expressed in means ± standard deviation. Statistical differences between treatments were evaluated by two-way analysis of variance (ANOVA) available in the software Graph pad Prism (9.1.0 (GraphPad Software, Inc., San Diego, CA, USA). Duncan’s multiple range test was used for mean separation and to define significant differences (*p* < 0.05). Figures were created using Origin Pro 8.0 software (OriginLab Corporation, Northampton, MA, USA).

## 3. Results and Discussion

### 3.1. Release of Anthraquinones during In Vitro Digestion

Figure 1 illustrates the release behavior of aloin (a) and aloe-emodin (b) in oral, gastric, and intestinal phases from spray-dried casein micelles loaded with anthraquinones extracted from the aloe vera plant. Aloin and aloe-emodin are two major anthraquinones found in the aloe vera plant. A significant difference (*p* > 0.05) can be seen in the release profiles of aloin and aloe-emodin from CMAQP, CMFDP, and CMWLAG in the oral phase by the action of α−Amylase enzyme. A lower release percentage of aloin (~4%) and aloe-emodin (~2%) was observed for casein micelles loaded with anthraquinones extracted from aloe vera plant (CMAQP), followed by ~7% release rate of aloin and 3% of aloe-emodin from casein micelles loaded with freeze-dried powder of aloe vera (CMFDP). In contrast to microcapsules of CMAQP and CMFDP, the release of aloin and aloe-emodin from CMWLAG was approximately the same for both, but there was far greater release of anthraquinones during the oral digestion phase (45%). This release of anthraquinones is not through the breakdown of casein micelles, indicating the surface attachment of anthraquinones that have not been encapsulated into the core after spray drying of CMWLAG. Furthermore, almost all the remaining anthraquinones (~90%) in CMWLAG were detected within the first 60 min of gastric digestion and only 15% and 10% of aloin and aloe-emodin were detected after 120 min of gastric digestion, while CMAQP released only half of that amount, indicating the high efficiency of encapsulation of anthraquinones. Although CMFDP showed a slightly higher release of aloin than CMAQP during oral digestion, the release amounts were far less than CMWLAG. This trend was followed by the release of aloe-emodin as well. For instance, the release of aloin and aloe-emodin after 60 min of gastric digestion was 30% and 25%, respectively, for CMAQP, whereas, after 120 min, the release was 37% and 35%, respectively. Regarding CMFDP, the release percentages of aloin and aloe-emodin were 35% and 28% after 60 min and 39% and 38%, respectively, after 120 min of gastric digestion.

These results align with our previous study [20], where CMAQP microcapsules followed by CMFDP microcapsules exhibited the highest encapsulation efficiencies for both aloin and aloe-emodin as opposed to CMWLAG. A significant portion of aloin and aloe-emodin was attached to the surface of the CMWLAG microcapsules, especially to the hydrophilic terminal of the casein micelles, instead of entrapping within the hydrophobic core [16] during the encapsulation process, and these surface-attached anthraquinones are easily released during simulated oral digestion. The pH-induced aggregation of casein micelles occurred during the encapsulation process with CMWLAG, creating a weaker, hollow, and branched casein micelle structure that did not tolerate spray drying to protect the core materials [16]. Hence, it can be determined that the long dissolution time of CMAQP and CMFDP would benefit the retention of the original powder structure, which made it difficult for digestive juices to penetrate the physical barrier and thus hindered the escape of anthraquinones, which can be verified by SEM images in Figure 2(D2,D3) as well.

Both CMAQP and CMFDP released anthraquinones significantly during SIF digestion. However, CMAQP showed 48% release within the first hour, followed by another 52% release for aloin, while CMFDP released 49% and 42% during the first and second hours of SIF digestion, respectively. In contrast, aloe-emodin showed a much higher release within the first hour of SIF digestion, irrespective of CMAQP or CMFDP, as compared to aloin release. CMFDP showed a significantly higher release of aloe-emodin during the first hour compared to CMAQP, indicating the efficiency of encapsulation by CMAQP. The hydrolysis and the relaxation of casein micelles [24] result in the disintegration of solid curds and the substantial release of anthraquinones. However, CMAQP delayed the diffusion process by protecting the anthraquinones within the core of CM due to strong hydrophobic interactions [25] and exerting a slower release of anthraquinones. In contrast, CMWLAG demonstrated an entirely changed mechanism during intestinal digestion, where anthraquinones were no longer released, as all had been released and degraded in the early phases (SSF, SGF). This means they were not protected by the casein micelles and their unstable nature caused significant loss during the initial part of in vitro digestion.

Overall, the release of encapsulated anthraquinones (aloin, aloe-emodin) from powdered microcapsules during in vitro digestion depended on various aspects, such as the casein micelles’ structure after spray drying, powder solubility, interactions between anthraquinones and casein micelles, and access and resistance of casein micelles to gastric and intestinal enzymes. In the small intestine, the non-covalent bonding of casein micelles and anthraquinones resulted in the dispersion of precipitates when mixed with SIF, indicating that the anthraquinones had been incorporated into the mixed micelles formed by the bile salts, making it possible for the CM to be absorbed into the intestinal epithelium [26]. So, the controlled release of anthraquinone could be achieved by CMAQP-based delivery systems. Regarding CMFDP, the microcapsules are efficiently rehydrated during gastric digestion, resulting in more anthraquinones being leaked into the supernatant. This might be due to the solubilization of colloidal calcium phosphate and exposure of the hydrophobic core of casein micelles that happened during spray drying [20]. However, in the small intestine, the hydrolysis of casein micelles due to the trypsin enzyme resulted in the explosive release of anthraquinones. 

### 3.2. SEM Images of Microcapsules during In Vitro Digestion

Figure 2(B2–B4) micrographs show different surface structures for AQP, FDP, and WLAG, which were used to incorporate the powders into the casein micelles. AQP showed a three-dimensional pentagon structure that has not been reported previously. However, bioactive components obtained from aloe vera skin showed some resemblance to the previous literature [27]. FDP showed a morphology with thin walls intersecting a three-dimensional array that repeats along the spongy surface with some holes and cracks. This aligned with previous work by [28,29]. In contrast, the WLAG structure showed the cleanest surfaces and well-maintained inner structure with rolled-out cell walls without shrinkage. 

Figure 2(B1–B4) corresponds to the morphology of spray-dried controlled casein micelles and microcapsules of CMAQP, CMFDP, and CMWLAG, respectively. Generally, CMAQP (Figure 2(B2)) displayed a wrinkled and non-spherical shape, corresponding to the fact that anthraquinones are bound to casein and embedded in the hydrophobic position of CMs after encapsulation. A decrease in the protein-to-mineral ratio caused the casein micelle to loosen and swell in CMFDP (Figure 2(B3)) with rough, fractal surfaces. However, CMWLAG (Figure 2(B4)) exhibited a cluster of spherical, smooth particles corresponding to aloe vera produced with randomly growing branches in stilliform geometry, where most of the CMs were aggregated. The reason is that casein micelles were aggregated during the encapsulation process by acidification. When spray-dried, the aggregated casein micelles enlarge in size, which causes them to loosen and swell. As a result of pH, salt content, and dehydration stresses, hydrophobic areas are exposed during spray drying, resulting in a loss of structural integrity. A detailed mechanism has been given in our previously published article [20].

Figure 2(C1–C4) correspond to encapsulated casein micelle microcapsules during oral digestion. Controlled CM showed smaller casein micelles in the oral phase than before digestion. Although alpha amylase does not directly affect casein micelles, it can affect them indirectly under certain conditions, such as changes in pH and temperature. Here a change in pH takes place, as the controlled casein micelles had a pH of 7.4 but dropped to 6.7 during oral digestion, which may have caused a decrease in CM size [30]. It can also be a consequence of mineral solubilization from casein micelles due to this decrease in pH. Partial demineralization of colloidal calcium phosphate (CCP) causes CM to shrink [31]. As the pH decreased, the voluminosity decreased due to ionization regression, resulting in a decrease in negatively charged proteins, thus reducing repulsive forces between adjacent chains. As a result, there was a progressive collapse of the outer hairy layer in milk at pH 6.6 [32]. However, it can be concluded that CMAQP in oral digestion showed a compact and dense structure, corresponding to the fact that casein micelles loaded with anthraquinone powder do not change in the oral phase. Casein micelles are compact microcapsules holding anthraquinones inside, having low dispersibility in the mouth for CMAQP. CMFDP showed a porous and rough structure during oral digestion, corresponding to a slight dispersibility of microcapsules. CMWLAG showed a modified structure compared to controlled CM and other microcapsules due to the precipitation of CM in the liquid phase through acidification during preparation, as stated in the previous section and previously [16]. 

The controlled casein micelles maintained their smooth surface and morphology with shuriken shapes, as reported previously [33] during gastric digestion after 60 min (Figure 2(D1)). This indicates that the casein micelles could resist dissolution even in highly acidic conditions, suggesting that they are highly stable and could remain intact for long periods [6]. In contrast to casein micelles, CMAQP and CMFDP (Figure 2(D2,D3)) microcapsules developed into heavily compact protein strands when exposed to the stomach’s acidic environment for 60 min [34]. In CMAQP and CMFDP, the CM structure was not corroded upon pepsin-induced coagulation [31], where the dense curd showed physical stability. This also suggested that exposure to the gastric environment causes a rapid restructuring of the microcapsules, resulting in a more tightly compacted protein structure. However, this has not been happening with the controlled casein micelles. The reason is that CMAQP and CMFDP microcapsules have been produced by reassembling casein micelles, as their structure has been reassembled after spray drying by exposing hydrophobic cores and the folding and unfolding of the CM structure [20]. These results are consistent with the previous literature [35], which demonstrated that rennet-induced casein micelle gels undergo large microstructural modifications in gastric acid, resulting in protein aggregation and coagulum formation [36]. These SEM images of CMAQP and CMFDP are consistent with the previous section, where the slower release of anthraquinones could be observed during 60 min of gastric digestion. The reason is that anthraquinones trapped in curded casein micelles demonstrated the ability of the CM curd structure to prevent anthraquinones from diffusing during in vitro digestion. Accordingly, curds provided a protective barrier for anthraquinones, preventing them from releasing until they were broken down and digested. Concerning CMWLAG (Figure 2(D4)), it was obvious that the spray-dried microcapsule structure had already been destroyed during the oral phase. The dissolved CMWLAG continued to undergo restructuring and dispersed in gastric digestion, leading to acid-induced destabilization of CM due to the acidic pH of the stomach, which could be seen at 60 min, as per SEM images. 

With an increase of one hour within the gastric phase, controlled CM-120 min showed a loose spherical structure compared to the original before digestion. Prolonged exposure of casein micelles to acidic conditions can cause individual caseins (α, β, and κ-caseins) to be released into the gastric phase. CM disruption might occur by the dissolution of calcium phosphate ions and a reduction in electrostatic repulsion between the casein proteins might occur. Hence, the collapse of hairy κ-casein ultimately reduces the steric stability of casein micelles [6], which corresponds to the images where disrupted CM could be seen attached to loosened CMs [37]. The effect of the acidic gastric environment is most clearly observed in CMAQP and CMFDP digesta, characterized by large and thick aggregate clusters, larger pores, and a higher area density compared to the first 60 min of digestion (Figure 2(E2,E3)). The increased degree of porosity of CMAQP and CMFDP curd-like structure after 120 min of gastric digestion is due to the prolonged exposure to low-pH conditions. The enzymatic hydrolysis of the curd is caused by gastric enzymes, particularly pepsin, which breaks down exposed CMs into smaller peptides, resulting in increased porosity [38]. In CMWLG, the consistency of the coagulum is more fragmented, with a more porous and aggregated structure after 120 min of gastric digestion. This differed from 60 min of gastric digestion, where no coagulum was seen but destabilized CMs [39]. However, prolonged exposure to stomach pH resulted in acid-induced coagulation [40], where individual casein proteins underwent coagulation and formed a curd-like soft structure (Figure 2(E4)).

Figure 2(F1–F4,G1–G4) showed morphologies of CMAQP, CMFDP, and CMWLAG after 180 and 240 min of intestinal digestion, respectively. From the controlled CM (Figure 2(F1)), it can be seen that when entering the intestinal phase, the casein micelle coagulum started to disintegrate into small particles and shrunk into brain-shaped structures after 180 min of SIF. The reason behind this is that an enzyme in the intestine (trypsin) hydrolyzed the casein micelles into smaller peptides and amino acids [41], which resulted in the loss of the CM’s original structure, and the CMs reassembled, as illustrated by the image (Figure 2(F1)). Moreover, a sudden increase in pH from the stomach (pH 3.0) to the intestines (6.7) tends to lead to reassociation of the casein proteins, thereby decreasing their porosity [42] as they form a more stable, shrunken structure during the 180 min of the intestinal phase.

Large curds disappeared for CMAQP (Figure 2(F2)) and CMFDP (Figure 2(F3)) after 180 min of intestinal digestion. However, a defined network was observed. During the first 60 min of intestinal digestion, it is possible for the casein micelle microcapsules to undergo rearrangement and reorganization as a result of trypsin action and pH change. Enzyme trypsin cleaves the amino acids (lysine, arginine) of the microcapsules CMAQP and CMFDP. These amino acids interact with the remaining intact caseins and anthraquinones, leading to a well-structured network. These results are consistent with the previous literature [18], where anthocyanins were encapsulated into casein micelles and well-structured CMs were seen after 120 min of intestinal digestion. Regarding CMWLAG-180 min of intestinal digestion, enzyme trypsin caused the degradation of CM structure. As reported previously, SEM morphology did not show any traces of casein micelles, although the morphological characteristics are similar to freeze-dried aloe vera gels. 

Controlled CM exhibits an interesting, elongated rod-shaped morphology after 240 min of intestinal digestion due to long exposure to alkali and trypsin enzymes, which caused the CM’s destruction into individual caseins without CCP; they aligned together into a partially unfolded rod-shaped structure. A rod-like shape was observed for C.N. particles without CCP as per SEC-SAXS data of caseinates (C.N.) [43]. It is the first time rod-shaped casein micelles have been reported after 240 min of intestinal digestion. It can be concluded that if intestinal pH is prolonged and trypsin enzyme is present, further hydrolysis, disruption of micellar integrity, loss of colloidal stability, formation of smaller structures, and alteration of casein micelles takes place to varying degrees based on the nature of the original microcapsule [44]. However, CMAQP, CMFDP, and CMWLAG morphological characteristics corresponded to the disintegration of CM micelles at 240 min of intestinal digestion. The reason is the continued hydrolysis of casein micelle microcapsules that the trypsin enzyme has restructured. Hydrolysis enhances the breakdown of individual casein proteins and peptide bonds, which release individual amino acids and ultimately lead to the gradual degradation of micellar structures [45].

### 3.3. Structural Modification of Microcapsules during In Vitro Digestion

The native form of proteins is converted to more unfolded forms during digestion. This conformational change is reflected by the Amide III band with changes in the frequencies of the N-H bending, C-N stretching, and C-C stretching vibrations [46]. These changes can provide insight into how the protein is being digested and how its structure is affected by the digestive process [47]. The most helpful absorption region for the infrared spectroscopic analysis of the secondary structure of proteins is the amide I, amide II and amide III regions [48]. Thus, the structural changes as a function of digestion were investigated through FTIR (Figure 3A–C). CM exhibited absorption bands at specific wavenumbers, including 3297 cm^−1^ (amide A), 3064 cm^−1^ (amide B), 1630–1660 cm^−1^ (amide I), 1517–1526 cm^−1^ (amide II), 1200–1350 cm^−1^ (amide III), and 1456 cm^−1^, as well as two weak bands at 1162 cm^−1^. These bands correspond to the turn structure of CM and are representative of stretching and deformation vibrations. 

As for Figure 3A, CMAQP microcapsules before digestion (CMAQPBD) showed the typical features of casein micelles with strong peaks of amide I and amide II bands at 1646 cm^−1^ and 1537 cm^−1^, respectively. This is in agreement with the previous literature [20]. During oral digestion of CMAQP, these FTIR peaks were inverted due to the modification of casein micelles as a result of a change in moisture content, solvation [49], and interactions with α-amylase action. FTIR spectra of CMFDP before digestion (CMFDPBD) (Figure 3B) showed the absence of secondary structure for amide I, but a lower-intensity amide III band was visible at 1243 cm^−1^.

A peak at 1243 cm^−1^ (Figure 3A) in microcapsules (CMAQPSSF, CMAQPSGF-1h, CMAQPSGF-2h, CMFDPSSF, CMFDPSGF-1h, CMFDPSFF-2h, CMAQPSGH-2h) corresponds to the amide III bands, which arose from vibrations in the peptide backbone caused by bending of N-H bonds, stretching of C-N bonds, and stretching of C-C bonds [50]. During CMAQPSSF and CMFDPSSF, the appearance of the amide III band is mainly due to the change in the hydrogen bonding pattern that links amino acids together [51]. However, the increased intensity of the amide III band in CMAQPSGF,1h and CMFDPSGF-1h was due to the release of some calcium phosphate ions and partial destruction of casein micelles. The lower intensities of amide III bands in CMAQPSGF-2h and CMFDPSGF-2h were due to a kink in the peptide backbone at Proline amino acid, which affects the peptide bond’s vibrational characteristics. The amide III bands in CMAQPSIF-1h, CMAQPSIF-2h, CMFDPSIF-1h, and CMFDPSIF-2h (1243 cm^−1^) were sharp and more prominent, which corresponded to the ongoing hydrolysis of CM under the influence of pH and trypsin enzyme at amino acid Glutamine. The CMWLAG (Figure 3C) exhibited similar peaks with similar intensities, unlike CMWLAGBD and CMWLAGSSF. During the oral phase of CMWLAG, all peaks were inverted upward due to the action of enzyme α-amylase and a drop in pH from 7.4 to 6.7, as described in the above sections. However, lower intensities of the characteristic peaks of the amide III band (1234 cm^−1^) and amide II band (1452 cm^−1)^ corresponded to the loss of casein micelle structure, which was destroyed during the oral and gastric phases, as described previously.

**Figure 3 foods-12-02844-f003:**
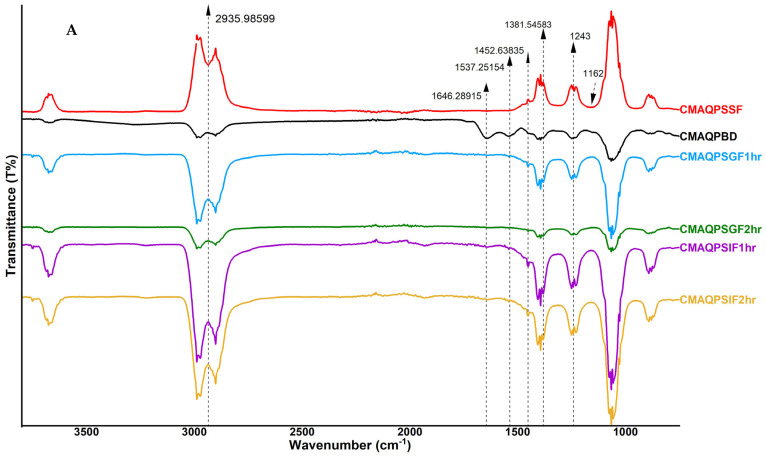
The FTIR spectra of microparticles before digestion (B.D.) and during in vitro digestion. (**A**) FTIR of CMAQP; (**B**) FTIR of CMFDP; (**C**) FTIR of CMWLAG.

### 3.4. Mechanism

A clear explanation of how anthraquinones are released during in vitro digestion of CMFDP, CMAQP, and CMWLAG is shown in Figure 4. A longer dissolution process was observed in the stomach when microcapsules with intact micellar structure (CMAQP, CMFDP) formed large curds and hindered the release of anthraquinones. However, the curds no longer existed after entering the small intestine. The SEM images of CMAQP and CMFDP verified these phenomena during gastric and intestinal digestion. FTIR peaks also revealed the decrease in the amide III bands. Their structure collapsed slowly shortly after that, releasing anthraquinones rapidly. The spray-dried CMWLAG microcapsules increased the solubility of casein micelles through the dissociation of CCP before spray drying and decreased the curdling ability. This, in turn, led to increased amounts of surface anthraquinones that had not been encapsulated to the core of the CM, which were then released during oral and gastric digestion phases. Following the entry into the small intestine, CMWLAG undergoes rapid hydrolysis and a significant structural change, as predicted by SEM images and FTIR results. Thus, the present study highlighted the suitability of CMAQP microcapsules for the controlled release of anthraquinones from aloe vera. 

**Figure 4 foods-12-02844-f004:**
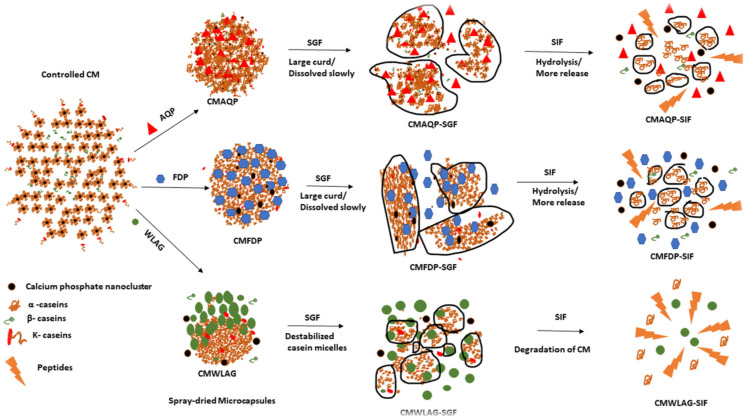
Mechanism of release behavior of anthraquinones from microcapsules of casein micelles (CMAQP, CMFDP, CMWLAG).

## 4. Conclusions

Successful encapsulation of anthraquinones in casein micelles is possible, but their stability and release behavior during in vitro digestion depends upon the type of aloe vera sample being encapsulated. Each type of microcapsule exhibited a different protection mechanism. CMAQP was the best-suited, followed by CMFDP. However, CMWLAG showed the least favorable controlled release of anthraquinones. Microcapsules of CMAQP and CMFDP showed poor solubility of CMs under initial digestion conditions and the formation of large curds prevented the release of anthraquinones in oral and gastric phases. The present study clearly demonstrated the suitability of utilizing casein micelles as a food-grade wall material for microparticles to encapsulate anthraquinones from aloe vera to improve their stability and bioavailability. However, careful consideration should be given to the nature and form of the anthraquinone that will be incorporated into the CM matrix. The results provide new insights into the development of functional foods and nutraceuticals enriched with bioactive compounds, such as anthraquinones from aloe vera. However, further studies could be conducted to investigate the size and structure of casein micelles with the different extracts and under different digestion conditions in solution form using scattering techniques, like DLS and SAXS.

## Figures and Tables

**Figure 1 foods-12-02844-f001:**
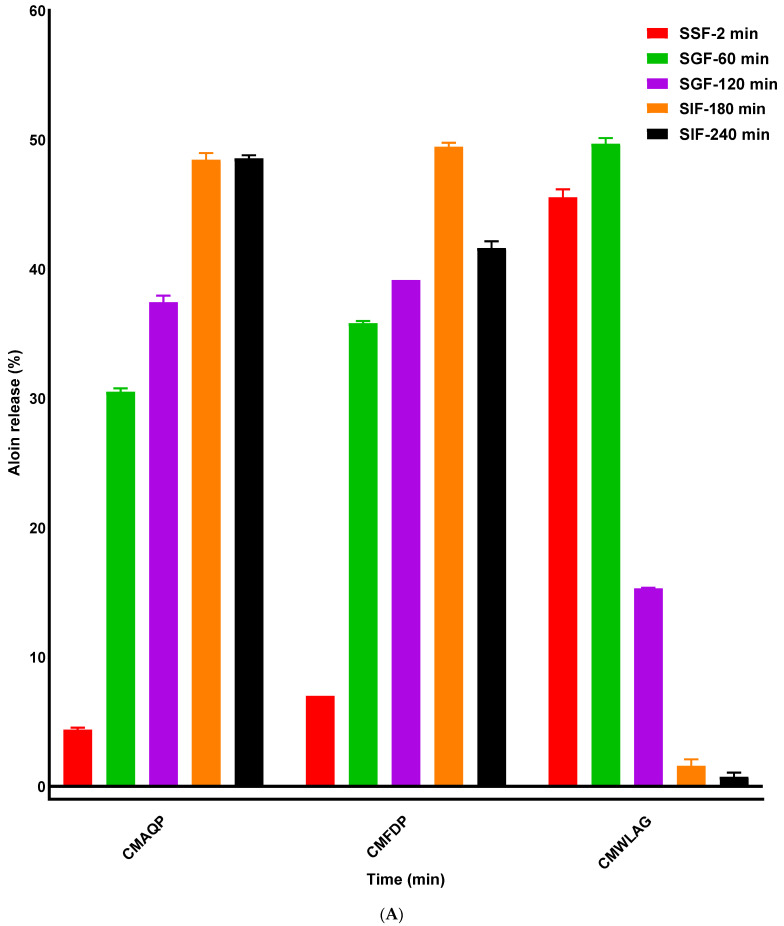
Release percentages of aloin (**A**) and aloe-emodin (**B**) from casein micelle microcapsules during in vitro digestion.

**Figure 2 foods-12-02844-f002:**
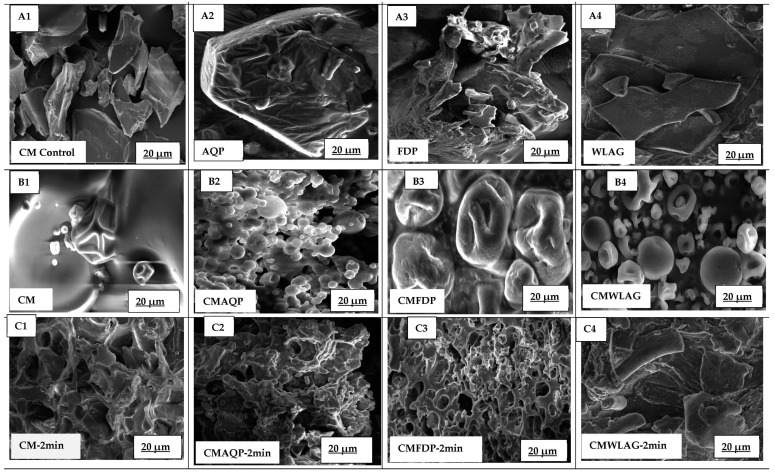
The changes in the morphology of spray-dried microcapsules during in vitro digestion. SEM images of controlled CM, controlled AQP, controlled FDP, and controlled WLAG (**A1**–**A4**) and SEM images of controlled CM (**B1**), casein micelle microcapsules of CMAQP (**B2**), CMFDP (**B3**), and CMWLAG (**B4**) before digestion. Casein micelle microcapsules of controlled CM (**C1**), CMAQP (**C2**), CMFDP (**C3**), and CMWLAG (**C4**) during oral phase. Microcapsules of controlled CM, CMAQP, CMFDP, and CMWLAG during gastric digestion for 60 min (**D1**–**D4**), during gastric phase for 120 min (**E1**–**E4**), during intestinal phase for 180 min (**F1**–**F4**), and intestinal phase for 240 min (**G1**–**G4**), respectively. Photographs were originally taken using a magnification of 5000× and the bar 20 μm scale.

**Table 1 foods-12-02844-t001:** Recommended concentrations in mL/L of electrolytes in simulated salivary fluid (SSF), simulated gastric fluid (SGF), and simulated intestinal fluid (SIF), based on human in vivo data.

Constituent (mL/L)	SSF	SGF	SIF
KCL	15.1	8.6	8.5
KH2PO4	4.6	1.12	1
NaHCO3	8.5	15.62	53.12
MgCl2	0.62	0.5	1.37
(NH4)2CO3	0.075	0.62	0
NaCl	0	14.75	12
HCL	as per pH adjustments	as per pH adjustments	as per pH adjustments
NaOH	as per pH adjustments	as per pH adjustments	as per pH adjustments

## Data Availability

The data presented in this study are available on request from the corresponding author.

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
