# Peer review of "The Release Behavior of Anthraquinones Encapsulated into Casein Micelles during In Vitro Digestion"

_foods, 2023, doi:10.3390/foods12152844_

Round 1

Reviewer 1 Report

The manuscript describes the encapsulation and release of anthraquinones in casein micelles under different conditions. The text is mostly clear and concise and may be of interest to the audience reached by the journal. Therefore, I'm in favor of some minor revisions, as stated below.   - On page 1, line 43, authors state that previous studies report the encapsulation efficiency of casein micelles, but no reference is cited. - In topic 2.2.1, the concentrations of aloe vera extracts mixed with casein micelle solutions are 20, 20 and 4 mg/mL for the different extracts. How did the authors choose these concentrations and why is the last one significantly lower than the other two? - On page 3, Table 1, the formulas for the salts have to be corrected (NaCl instead of NaCL and MgCl2 instead of Mgcl2, for example). - What kind of detector was used in HPLC to determine the anthraquinones concentrations? UV-Vis? - Were samples mixed with KBr in FTIR measurements? - It is not clear the concept of "colloidal calcium phosphate" that appears in some parts of the text. Is this any kind of particulate structure that composes the casein micelles? - Scale bars are missing in all images of Figure 2. - The FTIR spectra for the first compound in Figure 3 (red line) have the peaks outwards the 100% transmittance (that is peaks pointing to the upper side), while all the other spectra are pointing to the other side. Please, correct that. - As a suggestion of further investigation, the size and structure of casein micelles with the different extracts and under different digestions conditions could be studied in solution using scattering techniques, like DLS and SAXS.

Reviewer 2 Report

The manuscript entitled: The release behavior of anthraquinones encapsulated into casein micelles during in vitro digestion is interesting to scientific readers.

However I have few minor suggestions: The introduction should be improved and extended.

Table1 is poor and should be formatted. Figures are too big.

Minor editing of English language required

Reviewer 3 Report

The work presented by the authors is very interesting, the research is innovative and brings a lot of information that can ennoble the specialized literature about antraderivatives, namely Aloe and Aloe gel.

I agree with the publication, with a small observation at the end of the report, and the justifications are:

- The introduction systematizes the information about Aloe vera with a special report on the pharmacotherapeutic profile and the advantages of the microencapsulation of anthraquinone derivatives and the advantages of this processing, with the scoring of the already existing in vivo digestive models for different types of natural compounds;

- Materials and methods are properly described; the experimental design referring to Ultrasonic-assisted encapsulation of anthraquinones into CM and preparation of spray-dried microcapsules, In vitro static digestion, Release behavior of Anthraquinones during in vitro digestion, Scanning Electron Microscopy during in vitro digestion, Fourier transform infrared (FTIR) spectroscopy during in vitro digestion and Statistical Analysis;

- I guess the Results section also includes the Discussions; I believe that they are presented in detail on each sub-stage of the research, thus, the authors establish the percentage of release of aloin and aloemodine from the microcapsules with casein micelles, the changes in the morphology of the microcapsules, the analysis of the microparticles before and after digestion; in this section the authors present supporting data systematized in tables, figures or diagrams, graphs, photomicrographs; the authors use modern methods of analysis to achieve these objectives;

- The conclusions are very clear, very precise and are consistent with the objectives of the proposed study;

- The bibliography is justifiable.

It is a very interesting article, well argued, interesting and with very wide applicability.

THE AUTHORS PLEASE SPECIFY IN POINT 3 RESULTS AND DISCUSSION
